# Topical chlorhexidine 0.2% versus topical natamycin 5% for fungal keratitis in Nepal: rationale and design of a randomised controlled non-inferiority trial

Jeremy John Hoffman ![ORCID],[1,2] Reena Yadav,[2] Sandip Das Sanyam ![ORCID],[2] Pankaj Chaudhary,[2] Abhishek Roshan,[2] Sanjay Kumar Singh,[3] Simon Arunga,[1,4] Einoti Matayan,[5] David Macleod,[6] Helen Anne Weiss ![ORCID],[6] Astrid Leck,[1] Victor Hu,[1] Matthew J Burton[1,7]

For numbered affiliations see end of article.

**Correspondence to**
Dr Jeremy John Hoffman;
jeremy.hoffman@lshtm.ac.uk

## ABSTRACT

**Introduction** Fungal infections of the cornea, fungal keratitis (FK), are challenging to treat. Current topical antifungals are not always effective and are often unavailable, particularly in low-income and middle-income countries where most cases occur. Topical natamycin 5% is usually first-line treatment, however, even when treated intensively, infections may progress to perforation of the eye in around a quarter of cases. Alternative antifungal medications are needed to treat this blinding disease. Chlorhexidine is an antiseptic agent with antibacterial and antifungal properties. Previous pilot studies suggest that topical chlorhexidine 0.2% compares favourably with topical natamycin. Full-scale randomised controlled trials (RCTs) of topical chlorhexidine 0.2% are warranted to answer this question definitively.

**Methods and analysis** We will test the hypothesis that topical chlorhexidine 0.2% is non-inferior to topical natamycin 5% in a two-arm, single-masked RCT. Participants are adults with FK presenting to a tertiary ophthalmic hospital in Nepal. Baseline assessment includes history, examination, photography, in vivo confocal microscopy and cornea scrapes for microbiology. Participants will be randomised to alternative topical antifungal treatments (topical chlorhexidine 0.2% and topical natamycin 5%; 1:1 ratio, 2–6 random block size). Patients are reviewed at day 2, day 7 (with reculture), day 14, day 21, month 2 and month 3. The primary outcome is the best spectacle corrected visual acuity (BSCVA) at 3 months. Primary analysis (intention to treat) will be by linear regression, with treatment arm and baseline BSCVA prespecified covariates. Secondary outcomes include epithelial healing time, scar/infiltrate size, ulcer depth, hypopyon size, perforation and/or therapeutic penetrating keratoplasty (corneal transplant), positive reculture rate (day 7) and quality of life (EuroQol-5 dimensions, WHO/PBD-VF20, WHOQOL-BREF).

**Ethics and dissemination** The Nepal Health Research Council, the Nepal Department of Drug Administration and the London School of Hygiene and Tropical Medicine ethics committee have approved the trial. The results will be presented at local and international meetings and submitted to peer-reviewed journals for publication.

**Trial registration number** ISRCTN14332621; pre-results.

### Strengths and limitations of this study

► First large-scale randomised controlled clinical trial comparing chlorhexidine 0.2% to natamycin 5% for the treatment of fungal keratitis.

► This study benefits from a pragmatic design: as a non-inferiority trial, if chlorhexidine is found to be within a predefined non-inferiority margin of 0.15 logMAR of natamycin at 3 months a recommendation to use chlorhexidine 0.2% can be made; this is a far cheaper, easy to formulate medication that could significantly increase access to antifungal treatment for the target population.

► Clinicians are masked to the treatment allocation, however, due to different physical appearance it is not possible to mask patients to their allocated treatment.

► This study will also assess the superiority of either medication as well as a number of key secondary outcome measures, analysed by arm.

► First randomised controlled trial investigating fungal keratitis to use in vivo confocal microscopy as a diagnostic tool for the detection of fungal elements.

## INTRODUCTION

Fungal keratitis (FK) is a severe and potentially blinding corneal infection (figure 1).[1 2] The burden is greatest in tropical and subtropical countries, probably due to a combination of climate (higher temperatures and humidity) and frequent agriculture-related eye injuries.[3] It is one of the causes of microbial keratitis (MK) and accounts for between 20% and 60% of corneal infections diagnosed in



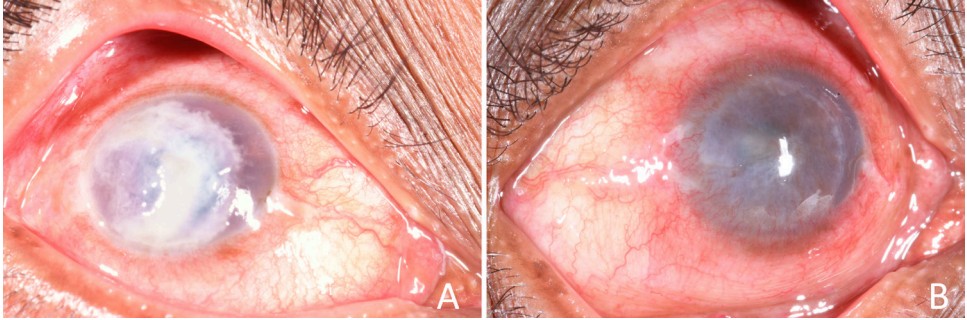

**Figure 1** Fungal keratitis and corneal scarring. (A) Active fungal keratitis with signs of acute inflammation and corneal ulceration. Photograph taken at presentation to SCEH. (B) Corneal scar, the blinding sequela of a resolved episode of fungal keratitis. Photograph taken at 2 months following presentation (same patient as (A)). SCEH, Sagarmatha Choudhary Eye Hospital.

tropical regions.[4] It is often inadequately treated with significant barriers to receiving appropriate, timely intervention, compounded by indiscriminate and inappropriate use of conventional medicines such as topical corticosteroids or harmful traditional eye medicines.[1 2 5] Furthermore, when appropriate treatment is available, up to 30% of patients receiving current 'gold-standard' therapy progress to corneal perforation and/or eye-loss (figure 2).[1 6 7]

Treatment for FK almost always involves topical antifungal agents. Surgical intervention, usually in the form of therapeutic penetrating keratoplasty (TPK), is generally reserved for cases of corneal perforation or progressive infection refractory to medical therapy. Corneal transplantation is also performed for visual rehabilitation after the acute infection has resolved. There are a limited number of antifungals available for treating FK, which fall into four main groups: imidazoles, triazoles, polyenes and fluorinated pyrimidines. These may be available topically, orally or by intravenous injection. Subconjunctival, corneal stromal or intracameral injections may also be given. The treatment of yeasts (*Candida spp*) is often different to filamentous fungi, with the former being more common in temperate climates and the latter in hot and humid locations.[8 9]

There have been several clinical trials comparing treatment options for FK, which have been systematically reviewed.[10 11] Natamycin (NATA), which was approved in the 1960s by the U.S. Food and Drug Administration (FDA) for FK, has been compared with a number of newer agents, including voriconazole. Natamycin and voriconazole have been compared in four trials, with the meta-analysis favouring natamycin.[6 10 12–14]

As a result, first line management of filamentous FK is usually with topical natamycin 5% when this is available. This was added to the WHO Essential Medicines List in 2017 for this indication. However, even when intensive topical natamycin is initiated, infections frequently progress relentlessly to perforation and loss of the eye in about a quarter of cases, figure 2.[1 6 7] Moreover, in many countries, antifungal eye-drop treatments are simply not available. This includes most countries in sub-Saharan Africa, some Asian countries and some countries in Europe.[1 2] Natamycin is relatively expensive even if it is available. Therefore, additional alternative and more affordable drugs are clearly needed if the outcome of these infections is to improve.

Chlorhexidine (CHX) is an antiseptic agent, with both antibacterial and antifungal properties. It is a widely used broad-spectrum biocide, killing micro-organisms through cell membrane disruption.[15–17] CHX has been used in ophthalmology for over 30 years as an eye-drop preservative and for sterilising contact lenses, and has also been used to treat *Acanthamoeba* and FK.[9 18–22] In a study of potential antifungal treatments, CHX was effective both in vitro against FK isolates from India and Ghana, as well as in an Indian case series.[23] Subsequently, two pilot randomised controlled trials (RCTs) of CHX for FK were conducted. In the first, three CHX concentrations (0.05%, 0.1%, 0.2%) were compared with each other

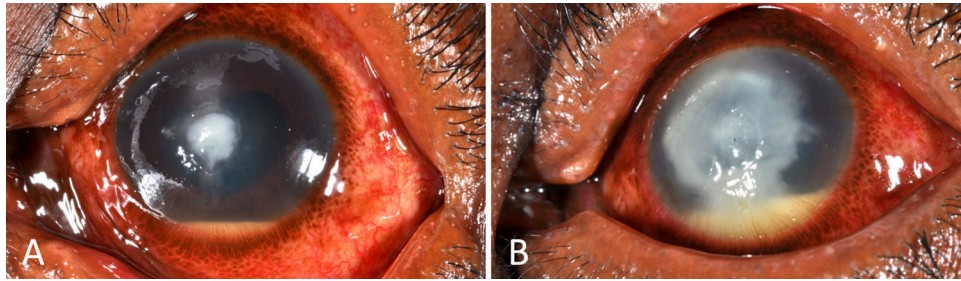

**Figure 2** Progressive fungal keratitis. (A) Early filamentous fungal keratitis; started immediately on intensive topical antifungal treatment (Natamycin 5%). (B) The same case 1 week later, unresponsive to intense natamycin 5% treatment, with progression of the infection.

and natamycin 5%; this concluded CHX 0.2% had reliable antifungal action.[19] The same concentration is used in mouthwash to prevent oral candidiasis. In the second trial, CHX 0.2% was compared with topical natamycin 2.5% (half standard concentration). There was evidence CHX produced a more favourable response by 5 days (RR 0.23, 95% CI 0.09 to 0.63).[18] A systematic review found a trend favouring CHX over natamycin in 'curing' by 21 days (RR 0.70, 95% CI 0.45 to 1.09), suggesting CHX might prove superior in adequately powered trials.[10] CHX is safe and well tolerated at these concentrations.[18 19 22 24] Based on this, CHX is used for treating FK in several countries.[1 9 11] However, the combined size of these two pilot trials comparing CHX and natamycin is not sufficient to reach firm conclusions. Currently, the meta-analysis indicates equipoise in terms of which treatment is the best for treating FK.

### Objectives

The primary objective of this study is to determine if topical CHX 0.2% is non-inferior to topical natamycin 5% for treating filamentous FK, in terms of vision at 3 months. The secondary objectives are: (1) to determine whether either treatment (CHX 0.2% or natamycin 5%) is superior to the other, in terms of vision at 3 months and (2) to determine whether there is a difference between CHX 0.2% and natamycin 5% in terms of secondary clinical outcomes including infiltrate/scar size, time to re-epithelialisation, reculture rates at 1 week and the effect of the alternative treatments on the Quality of Life of participants.

CHX is cheap, stable and easily prepared by aqueous dilution. If CHX is found to be non-inferior (or even superior) to natamycin this offers the potential of an effective, affordable and accessible treatment for FK, which could benefit millions of people each year who currently have no treatment options. This trial is a response to this expressed need from both clinicians and patients for a readily available and affordable medication for fungal infection.

### METHODS AND ANALYSIS
### Trial design

We are conducting a single-masked, non-inferiority RCT comparing CHX 0.2% to natamycin 5% for the treatment of FK. The non-inferiority design of this trial offers a clinically pragmatic way to address this important question: if CHX is found to be within the prespecified non-inferiority margin of 0.15 logMAR (about 1.5 Snellen lines) then CHX may prove to be a sustainable solution for this aspect of the complex problem of FK.

### Trial summary

This RCT follows a two-stage recruitment process (figure 3). All patients presenting with acute MK are reviewed and enrolled into stage 1, following written, informed consent. This involves history, examination

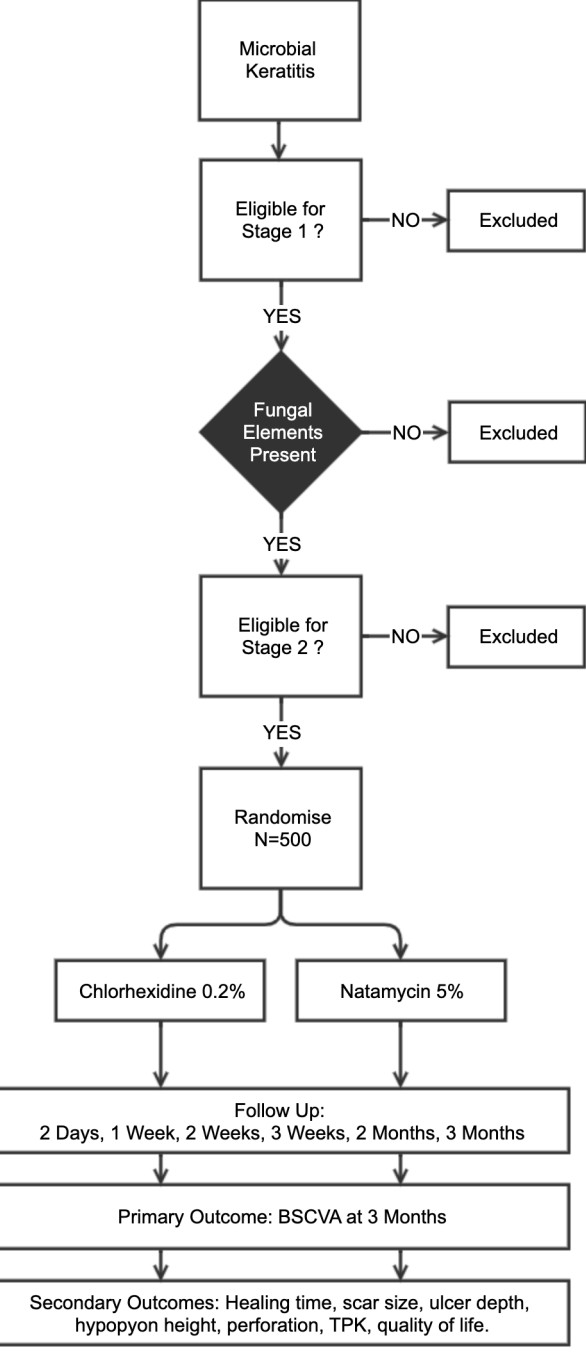

**Figure 3** Overview of the clinical trial. Microbial keratitis is defined as presence of corneal epithelial ulceration (>1 mm in diameter), corneal stromal infiltrate and signs of acute inflammation (eg, conjunctival injection, anterior chamber inflammatory cells, hypopyon). Fungal elements to be detected by smear microscopy and/or confocal microscopy. Those eligible will be randomised 1:1 to CHX or NatA (n=500). BSCVA, best spectacle corrected visual acuity; CHX, chlorhexidine; TPK, therapeutic penetrating keratoplasty.

and investigations (corneal scrapes for microbiological assessment and in vivo confocal microscopy (IVCM)). If there is evidence of fungal hyphae on smear or confocal microscopy, patients then proceed to stage 2. A trial eligibility checklist is completed and stage 2 written informed consent is conducted. We will recruit 500 patients into

stage 2. Eligible FK patients are then randomised 1:1 to receive either natamycin 5% or CHX 0.2% topical treatments hourly for the first week, then 2 hourly for the subsequent 2 weeks. Ongoing treatment duration will then be tailored to clinical response. Study personnel are masked to the treatment allocation. Patients are usually initially admitted and followed up on day 2, day 7 (with reculture), day 14, day 21, month 2 and month 3. The primary outcome is the best spectacle corrected visual acuity (BSCVA) at 3 months.

### Trial setting

Low-land Nepal, a region with a high burden of FK, provides a suitable location to conduct a clinical trial that requires a relatively large sample size. This trial will be conducted in Sagarmatha Choudhary Eye Hospital (SCEH), Lahan, Siraha District in south eastern Nepal. SCEH treats approximately 800 cases of keratitis per month, half of which is attributable to fungal infection.[1 25] SCEH is a tertiary-level ophthalmic hospital, with several satellite Eye Care Clinics (ECCs) that refer patients to SCEH directly as required. SCEH and its ECCs serve an estimated population of about 5 million people. Due to its proximity to the border, approximately 50% of the outpatients are Indian nationals. It is anticipated that the study participants will present to the hospital from multiple districts within the region. Potentially eligible individuals will be recruited from outpatient clinics or referred directly by the ECCs to the study team. Based on the numbers of patients attending, it should be possible to complete recruitment of 500 participants within 6–12 months. A second separate trial in East Africa (Tanzania and Uganda) will also be conducted to compare CHX 0.2% to natamycin 5% following a very similar protocol to the one described here. This will be registered as a separate trial. This will enable us to assess the generalisability of the findings in two geographically distinct regions with potentially different fungal aetiologies and susceptibility patterns.

### Eligibility criteria

Potential participants need to meet all the inclusion criteria and have none of the exclusion criteria listed in table 1. In summary, they need to have an active FK defined as acute MK characterised by corneal epithelial ulceration (>1 mm in diameter), corneal stromal infiltrate and signs of acute inflammation (eg, conjunctival injection, anterior chamber inflammatory cells, hypopyon) in conjunction with evidence of a filamentous fungal infection on smear microscopy and/or IVCM. There is strong evidence supporting the use of IVCM for diagnosing filamentary FK. Studies have reported sensitivities of 85.7%–89.2% and specificities of between 81.4% and 92.7%, respectively.[26–28] As some patients will be enrolled on the basis of the results of IVCM which is unable to detect most bacteria reliably, some patients with microscopically confirmed fungal infection will subsequently also be found to have had mixed infection at the time

of being recruited into the study, as bacterial cultures may become positive a few days after enrolment. Based on previous experience at SCEH this is likely to account for about 10% of cases. These patients are included in the study but excluded from the primary analysis of the primary outcome (see below). Secondary analyses will include mixed infections.

### Consent procedures

There are two independent consent stages in this trial: stage 1, where all adult patients with MK are eligible; and stage 2, only for FK patients meeting the eligibility criteria. The two-stage process enables data collection on all potential patients at baseline before a diagnosis of FK is confirmed. All patients who are eligible to participate will be given a participant information sheet in Nepali and its contents read to them. There will be an opportunity to discuss any questions that they might have. If the patient would like to participate, they will be asked to read and sign or place a thumb print on the study consent form. The consent will be witnessed by the eye health worker, confirmed by a signature on the form. For patients who are unable to read the documentation a second witness who is unrelated to the study is required. Consent forms are given in online supplementary appendix 1.

### Baseline assessment

The detailed baseline assessment is described in table 2. This includes clinical examination, corneal photography, IVCM and the collection of microbiology samples. Quality of life questionnaires will also be completed (EuroQol-5 dimensions (EQ-5D), World Health Organization Prevention of Blindness and Deafness 20-item Visual Functioning Questionnaire (WHO/PBD-VF20), World Health Organization Quality of Life: Brief Version (WHOQOL-BREF), details of the scoring for these are given in table 2.

### Randomisation and masking
#### Sequence generation

A computer-generated randomisation list will be prepared by an independent statistician at London School of Hygiene and Tropical Medicine (LSHTM), who will hold the sequence, will not be masked, and will not be involved in any other aspect of the trial. The sequence is in a 1:1 allocation ratio of CHX to NATA, with a random block size (2, 4 or 6).

#### Allocation concealment and implementation

The randomisation sequences will be concealed in sequentially numbered, opaque envelopes. The envelopes will be prepared by a person independent of all other aspects of the trial. The randomisation administrator (nurse or pharmacist) will conduct the random allocation procedure. The investigational products will be stored in the trial coordination office in a locked and separate drug cabinet dedicated for this clinical trial only. The cabinet will only be used for drug storage and will only be accessed by the randomisation administrator. The randomisation administrator will work in a separate room

**Table 1** Inclusion and exclusion criteria for enrolment in stage 1 (MK cases) and stage 2 (the randomised controlled trial)

| Inclusion criteria (all must be met) | Exclusion criteria (any of the following) |
|---|---|
| **Stage 1** | |
| 1. Acute MK characterised by: | 1. Patients aged less than 18 years |
| ▶ Corneal epithelial ulceration >1 mm diameter | 2. Patients unable or unwilling to provide informed consent |
| ▶ Corneal stromal infiltrate | 3. Patients who do not have acute MK or where there is a more likely alternative diagnosis |
| ▶ Acute inflammation: for example, conjunctival injection, anterior chamber inflammatory cells, hypopyon | |
| 2. Adults (18 years and older) | |
| 3. Able to provide informed consent | |
| **Stage 2** | |
| 1. Acute MK characterised by: | 1. Unwilling/unable to participate in trial and/or attend follow-up |
| ▶ Corneal epithelial ulceration >1 mm diameter | 2. Aged less than 18 years |
| ▶ Corneal stromal infiltrate | 3. Pregnancy: self-reported, or by urine pregnancy test if uncertain. |
| ▶ Acute inflammation: for example, conjunctival injection, anterior chamber inflammatory cells, hypopyon | 4. Breast feeding: self-reported |
| 2. Filamentous fungal hyphae visualised on smear microscopy and/or IVCM | 5. Prior topical antifungal treatment |
| 3. Agree to be randomised to either treatment arm and are able to give informed consent | 6. No light perception in the affected eye |
| 4. Agree to be followed up at 2 days, 1 week, 2 weeks, 3 weeks, 2 months and 3 months | 7. Fellow eye visual acuity <6/60 |
| 5. Adults (18 years and older) | 8. Acanthamoebic infection visualised by smear microscopy or IVCM |
| | 9. Clinical evidence of herpetic keratitis |
| | 10. Known allergy to study medication (including preservatives) |
| | 11. Previous keratoplasty in the affected eye |
| | 12. Bilateral corneal ulcers |
| | 13. Very severe ulcers warranting immediate evisceration or conjunctival flap |
| | 14. Endophthalmitis |

IVCM, in vivo confocal microscopy; MK, microbial keratitis.

in the clinic. After the patient has been recruited and assessed, they will be guided to this separate room where the randomisation allocation will be conducted.

### Masking

The topical treatments being compared in this trial have a different appearance: the CHX 0.2% is a clear, colourless solution and the natamycin 5% is an opaque, white suspension. Therefore, it is not possible to mask the participants to this difference in visual appearance. However, the patients will not be told which treatment they have been allocated. Prior to any follow-up clinical examinations, a nurse, otherwise uninvolved in the study, will wipe away any white natamycin residue from the patient's eyes to avoid unmasking the clinical assessor. This procedure was used successfully in other trials.[6] All clinicians involved in the clinical assessment of the patients will be masked

to the allocation. The statistician who will perform the primary analysis will be masked to the allocation and only receive the actual allocation code sequence from the independent statistician after the analysis code has been prepared and pre-tested with a test sequence. The primary outcome is visual acuity at 3 months assessed by an optometrist who will not have been involved in any other aspects of the trial and masked to the allocation. By 3 months, the treatment courses are likely to be completed. There will be masked grading of the photographs to independently confirm outcome measures and assess for any systematic bias on the part of the clinical examiners.

### Unmasking

Unmasking is a serious action and should only be performed if necessary to ensure the safety of a study participant. Anyone unmasked to randomised treatment

| Table 2 | Baseline assessment |
|---|---|
| **Assessment** | **Details** |
| Visual acuity | Presenting, Pin-Hole and best spectacle corrected visual acuity) will be measured using an ETDRS Tumbling-E logMAR 3 m chart (Good-Lite Inc, USA) mounted on an ESC 2000 ETDRS LED Cabinet, (Good-Lite Inc, USA) by a trial-certified optometrist, for each eye separately |
| Contrast sensitivity | Measured using the Peek Contrast Sensitivity smartphone application running on Android OS with a Sony Xperia Z3 Compact smartphone (Sony, Japan).[26] |
| Clinical photographs | Photographs will be taken separately of both corneas using a Nikon D7500 camera with an AF-S Micro Nikkor 105 mm lens and lens mounted SB-200 flash units (Nikon, Japan). A standardised photography protocol is used to ensure images can be compared between time points. Standardised magnification will be used to allow epithelial defect and stromal infiltrate size measurements to be made. |
| Slit-lamp examination | Both eyes will be examined using a slit-lamp biomicroscope (standard ophthalmology examination) to assess the anterior segment of the eye. This examination will be performed by an ophthalmic clinician experienced in managing MK. Particular attention will be paid to the following features: <br><br>1. Eyelids: trichiasis, lagophthalmos, facial weakness, Bell's reflex <br>2. Suppuration <br>3. Conjunctival inflammation <br>4. Corneal sensation <br>5. Cornea epithelial defect (measuring the longest dimension and the longest perpendicular) and ulcer depth <br>6. Corneal inflammatory infiltrate depth, size, profile, colour, edge pattern, texture, satellites <br>7. Anterior chamber inflammatory cells, hypopyon, endothelial plaque <br>8. Relative afferent pupillary defect |
| In vivo confocal microscopy (IVCM) | The Heidelberg Retinal Tomograph 3 IVCM enables the clinician to examine the cornea down to the cellular level. It is able to detect the presence of fungal hyphae.[27 28] A sterile, single-use disposable cap covers the objective lens and is changed between patients. Volume scans will be performed which provide a series of 400×400 μm images over a depth range of 80 μm. The resolution of the corneal scanning module is 7.6 μm. IVCM images will be collected in a systematic way, starting at the centre of the ulcer, then at the superior, inferior, nasal and temporal borders of the ulcer. Volume scans will be performed in all of these locations, starting at the level of the corneal epithelium, and ending at the deepest affected aspect of the cornea assessed from IVCM images. Images will be assessed during the examination. |
| Ocular sample collection | The following samples will be collected from the corneal ulcer of each patient at the baseline assessment: <br><br>1. Corneal scrape specimens for microscopy and microbiological culture. A corneal scrape will be collected from the corneal ulcer after application of preservative free proxymetacaine local anaesthetic eye-drops (Minims). Sterile needles are used to take corneal scrape specimens and then place on to glass slides for immediate Gram stain, KOH and Calcofluor white. Samples will be directly inoculated onto blood, chocolate, Sabouraud agar and broths for culture. <br><br>2. Corneal specimen collection for PCR. Two sterile swabs will be gently swept over the surface of the corneal ulcer and placed into a 2 mL tube. The swabs will be for pathogen detection by PCR, fungal sequencing and assessment of point of care tests for fungal infections. Swabs will be stored dry at −80°C. If swab yields are found to be too low for analysis an additional corneal scrape will be collected for PCR. The analysis of the PCR samples will not form part of the RCT workup and report. |
| HIV testing | All individuals presenting with MK would be offered counselling and testing services. If this is found to be positive and the patient is unknown to the HIV care services an appropriate referral will be made. HIV testing is performed using HIV Tri-Dot rapid diagnostic test (J. Mitra & Co, India) |
| Random blood glucose | There is a suggestion that individuals with diabetes may be more susceptible to FK. Participants will be offered a random blood glucose test, on a finger prick sample, analysed using HumaLyzer Primus (HUMAN Gesellschaft für Biochemica und Diagnostica mbH, Germany). If this is above 6.1 mmol/L they will be referred to the hospital physicians for assessment and formal diagnosis of impaired glucose tolerance or diabetes mellitus. This level is considered a suitable cut-off to detect individuals with diabetes and has been validated in a south-Asian population.[34] |

Continued

**Table 2** Continued

| Assessment | Details |
|---|---|
| Quality of life questionnaires | For those with confirmed FK and who are enrolled in the trial, there will be several additional baseline assessments to evaluate the impact of FK on quality of life. |
| | Vision-related quality of life (VRQoL): will be assessed by a vision disease specific tool the WHO/PBD-VF20.[35] This tool measures the impact of visual impairment in the person's life including mental well-being, dependency and social functioning. These have been used in a number of other vision related studies to show a difference in QoL.[36 37] This instrument consists of 20 questions divided into three sub-scales: visual symptom, general functioning and psychosocial. It begins by asking the patient *'Overall, how would you rate your eyesight using both eyes?'*; and uses a five point scale answer option such as *'very good'*, *'good'*, *'moderate'*, *'bad'*, *'very bad'*. The test is scored out of 100, with higher scores reflecting a better VRQoL. |
| | General health-related quality of life: We will use the EQ-5D questionnaire and EQ-Visual Analogue Scale. The EQ-5D is a standardised tool to measure health outcomes.[38] Patients will also be assessed using the WHOQOL-BREF.[39] This has good applicability in low and middle-income countries as it was developed simultaneously from concept across 18 countries in Africa, Asia and Latin America. It measures 4 domains of health: Physical Health, Psychological Health, Social Relationships, and Environment. It asks respondents 26 questions how much (frequency) they have experienced and/or were able to do things (eg, feel safe, able to concentrate, enjoy life) in the past 4 weeks and how satisfied they are with certain aspects of their lives (eg, sleep, capacity for work). Each question is scored between one and five in a positive direction, with one being attributed for a very low or dissatisfied quality of life, and five being very good or very satisfied with their quality of life (ie, higher scores denote a higher quality of life). Each domain has its own score calculated by calculating the mean of the item scores within each domain. In addition, there are two items that are examined separately: question 1 asks about an individual's overall perception of quality of life and question 2 asks about an individual's overall perception of their health. These are scored on the same positive scale from one to five. The mean domain scores can then be multiplied by four in order to make domain scores comparable with the scores used in the WHOQOL-100.[39] |

Assessment performed at baseline with details of how they are made.

AF-S, Autofocus Single; EQ-5D, EuroQol-5 dimensions; ETDRS, Early Treatment Diabetic Retinopathy Study; FK, fungal keratitis; KOH, Potassium Hydroxide; MK, microbial keratitis; RCT, randomised controlled trial.

for the purpose of creating analyses for independent data and safety monitoring committees will not be involved in any other aspects of the conduct or final analysis of the study. A list will be maintained of all unmasked members of staff and will be approved by the chief investigator. Such staff members must sign a document to indicate that they are aware of their responsibilities with respect to confidentiality. The processes used to provide access to unmasked treatment codes and reports relating to these codes shall be documented.

### Intervention and treatment

The standard of care in this hospital is for cases of FK to be admitted. If willing, patients will be admitted for close observation and supervised treatment, until there are signs of improvement and the supervising clinician considers it safe to provide ongoing management on an outpatient basis.

### Trial treatment arms

1. CHX 0.2% w/v eye-drops
   CHX 0.2% w/v solution, as eye-drops, will be applied to the surface of the infected eye (one drop per application). The CHX 0.2% w/v eye-drops used in these studies is produced by Mandeville Medicines, UK.
2. Natamycin 5% eye-drops
   Natamycin 5% w/v suspension, as eye-drops, will be applied to the surface of the infected eye (one drop per application). Topical natamycin 5% is produced by FDC Pharmaceuticals, India.

### Dosing schedule

Both trial treatment arms will follow the same dosing schedule. Eye-drops will be given hourly day and night for 48 hours, then hourly while awake for 5 days, and then 2 hourly while awake for two further weeks. If the ulcer has healed (epithelial defect less than 1 mm, infiltrate resolved, with or without corneal scarring), then treatment is stopped. If resolving stromal infiltration and/or epithelial defect >1 mm but <5 mm, treatment is reduced to four times daily. If resolving but with epithelial defect >5 mm and/or stromal infiltration/hypopyon, treatment is reduced to six times daily. Treatment duration will be tailored to clinical response, with patients reviewed regularly while on treatment in addition to their scheduled study visits.

### Topical treatments

Several other topical medicines are used in the standard care of people with corneal infections:

1. Fluorescein sodium ophthalmic strips (Contacare Ophthalmics and Diagnostics, India) to highlight the area of cornea epithelial defect.
2. Anaesthetic eye-drops to anaesthetise the cornea before procedures such as microbiology samples:

Proxymetacaine 0.5% eye-drop Minims (Bausch and Lomb, UK).

3. Antibiotic eye-drops may be used as per the judgement of the treating ophthalmologist if there is a suspicion of mixed bacterial and FK: Moxifloxacin 0.5% eye-drops (Centaur Pharmaceuticals, India).

4. Mydriatic eye-drops for pupil dilation to reduce discomfort from the infection: cyclopentolate 2% eye-drops (Aurolab, India), three times a day.

5. Ocular hypotensive eye-drops if the intraocular pressure is elevated >25 mm Hg. This will be at the discretion of the supervising clinician; usual first line treatment is timolol 0.5% eye-drops (Allergan, India).

### Ancillary treatment in patients refractory to trial medication

In patients with progressive FK despite 7 days or more of trial medication, additional treatment options will be considered and offered. After repeating microbiological tests to rule out a mixed bacterial infection, deteriorating patients are started on topical voriconazole 1% hourly if the ulcer is superficial only. For ulcers that are deeper than 75% of the corneal thickness, oral ketoconazole 200 mg two times a day is added (with monitoring of liver function). The choice of ancillary treatment is due to local availability. If, following 7 days of additional treatment, despite these measures there is ongoing progression, surgical management such as a TPK can be considered.

### Non-pharmacological treatment

It is sometimes necessary to perform surgical procedures during the management of corneal infections. In the context of the trial these will be performed by the supervising consultant ophthalmologist. These can include:

1. Insertion of a bandage contact lens for very small perforations.
2. Tissue glue and patch for small perforations.
3. Corneal transplant (TPK) for progressive fungal infections that are refractory to medical management or large perforations.
4. Conjunctival flaps to cover corneas in non-healing ulcers.

### Outcome measures
#### Primary outcome measure

The primary outcome will be BSCVA in logMAR units at 3 months. This will be measured by a trial-certified optometrist, independent of all other aspects of the study and masked to allocation. This has been the primary outcome of several other trials and will therefore facilitate comparison. Three months is the time at which clinical experience suggests most corneal ulcers have usually healed. BSCVA is chosen as it is easy to measure and is of functional significance. We will measure the BSCVA using an LED-backlit, Tumbling-E LogMAR chart (Good-Lite, Illinois, USA) under controlled conditions. We will also measure vision using Peek Acuity, a validated smartphone application, in situations where the patient is unable to attend the hospital for follow-up and outcome data is

only available through a domiciliary visit.[29] For patients who have counting fingers (CF) vision or less, predefined logMAR values will be assigned based on previous clinical trials.[30 31] Similarly, for patients who undergo a corneal transplant (TPK), a predefined visual acuity of 1.9 logMAR will be given, or last observation carried forward (whichever is the better vision), as per previous studies.[6]

#### Secondary outcome measures

We will be assessing a number of secondary outcome measures that relate to different measures of visual acuity such as Peek Acuity; clinical signs of healing such as reduction of epithelial defect; microbiological culture rates and other clinical outcome measures such as scar size or perforation rate. These secondary outcomes together with their analyses are outlined in the Analysis Plan section.

### Outcome assessments

Participants will be reassessed at 2 days, 1 week, 2 weeks, 3 weeks, 2 months and 3 months following enrolment. Additional examinations, outside the trial protocol schedule may be conducted by the supervising clinician as indicated. The specific assessments to be carried out at each visit are indicated in table 3. These will be conducted in the same way described for the baseline assessment as described (table 2). At each follow-up assessment, the participant will be asked about adherence to treatment and symptoms, including side effects, if still receiving trial treatment. This will be recorded in the case record file. To monitor trial medication adherence, the study participants will be asked to bring their eye-drop bottles to the 1-week, 2-week and 3-week follow-ups. The amount of remaining medication will be measured by weighing the bottle and will be compared with the anticipated remaining amount provided the drops were being used as instructed, resulting in a ratio of actually remaining to what is anticipated. Participants will be resupplied with medication as needed. The presenting visual acuity will be measured on each visit. In addition, the BSCVA will also be measured at the 3-month follow-up (primary outcome measure). At the final 3-month follow-up, the three quality of life questionnaires will be repeated. On each occasion the eyes will be examined using a slit-lamp and the cornea photographed. The IVCM will be repeated at 1, 2 and 3 weeks to determine whether there is evidence of fungal hyphae resolution with the fragmentation of linear elements. At the 1-week follow-up if the ulcer has not healed it will be rescraped for repeat culture to determine whether the keratitis is now culture positive or culture negative. We will give the study participants appointment cards for the next follow-up and they will be reminded about the follow-up a week prior to the date. Public transport costs will be paid for participants who are outpatients.

### Treatment review

At each follow-up visit, participants will be reviewed by an ophthalmic clinician experienced in the management of FK. Clinical responses to antifungal therapy tend to be relatively slow (compared with bacterial infections). Prolonged topical treatment courses of 4–6 weeks are usually needed.

**Table 3** Baseline and follow-up assessment components

| Assessment item | Baseline | Day 2 | Day 7 | Day 14 | Day 21 | Day 60 | Day 90 |
|---|---|---|---|---|---|---|---|
| History/baseline questionnaire | X | | | | | | |
| Check treatment adherence | | X | X | X | X | X | X |
| Check for side effects | | X | X | X | X | X | X |
| Visual acuity—presenting | X | X | X | X | X | X | X |
| Visual acuity—BSCVA | X | | | | | | X |
| Contrast sensitivity | X | | | | | | X |
| Slit-lamp examination | X | X | X | X | X | X | X |
| Cornea photography | X | X | X | X | X | X | X |
| In vivo confocal microscopy | X | | X | X | X | | |
| Cornea samples (microbiology/PCR) | X | | X | | | | |
| Quality of life tools | X | | | | | | X |

BSCVA, best spectacle corrected visual acuity.

Therefore, a change in therapy is usually not made for at least the first week. Other interventions may be indicated such as application of glue to corneal perforations, conjunctival flaps or TPK (corneal transplant).

## Stopping rules

If the study eye develops any serious adverse outcomes, the antifungal study medication may be discontinued if it is considered to be responsible for the adverse event. The patient will then be treated at the discretion of the supervising ophthalmologist, without breaking the randomisation code. If the medication is stopped, the patient will continue with the scheduled follow-up examinations.

## Lost to follow-up

Lost to follow-up rates are expected to be low, based on clinical experience. Participants who do not present for their follow-up visit will be contacted by telephone. Reasons for the lost to follow-up will be identified. Patients will be counselled about the importance of attending for ongoing treatment and monitoring. If they are unwell or unable to attend the hospital for some specific reason the study team will arrange to visit them in their home. Reasons for lost to follow-up will be recorded and reported.

## Data collection, management, confidentiality and access to data

Data will be collected using paper clinical record forms. These will be stored securely at SCEH and scanned electronic copies taken at the end of the day, stored on an encrypted drive with encrypted backups made daily both on and offsite. These data will be double entered into two separate MS Access databases. After double data entry has been completed data will be cleaned using EpiData V.3.1 software. Data entry is supervised by the local study coordinator on a daily basis, with data collection and data entry progress being reviewed by the study coordinator and chief investigator at LSHTM on a weekly basis. Data protection and confidentiality is maintained through restricted access to the database system and

cupboard where the paper documents are kept. Only authorised users will have access to the locked filing cabinet. The database will be password protected, with each data entry staff member having their own password. Data exports for further analysis will be anonymised.

## Data and safety monitoring board

The data and safety monitoring board (DSMB) for this trial includes independent experts in bioethics, biostatistics, epidemiology and ophthalmology appointed by the trial steering committee and approved by the national regulatory authorities, before the start of the study. The DSMB meets at least twice each year and organises teleconferences as needed for progress reporting. The study protocol and modifications are subject to review and approval by ethics committees in Nepal and LSHTM, and by the DSMB. The DSMB will monitor any severe or unexpected events and oversees the data collected. The DSMB will be responsible for reviewing the results of the interim analysis and determining whether or not the trial should continue, with or without modifications.

## Monitoring for harm

Patients will be monitored at each visit for adverse events or reactions. We will follow standardised definitions for adverse events, adverse reactions, unexpected adverse reactions, serious adverse events or reactions, and suspected unexpected serious adverse reaction. These, along with the reporting scheme, are given in online supplementary appendix 2.

## Biological specimens

The processing and analysis of biological specimens are detailed in online supplementary appendix 3.

## Sample size considerations

The study is powered to test the hypothesis that CHX is non-inferior to NATA in terms of the primary outcome (BSCVA at 3 months) at a prespecified non-inferiority margin (Delta) of 0.15 logMAR units. It is possible that CHX is non-inferior to NATA, given pilot RCT data that showed no statistically

**Table 4** Secondary outcome measures that will be investigated as part of the trial, together with analysis details

| Secondary outcome measure | Details |
|---|---|
| Three-week BSCVA | We will analyse the secondary outcome of 3 weeks BSCVA in logMAR in the same manner as the primary analysis of the primary outcome described above. The 3 weeks BSCVA will include values taken between 18 days and 5 weeks, with the value closest to 3 weeks used. |
| Presenting VA by Peek | We will analyse the presenting VA by Peek Acuity with and without pinhole at 3 months as a secondary outcome. This will be of interest if we are unable to get reliable BSCVA measurements at 3 months (ie, if patients fail to attend and we need to attend their houses for visual acuity testing). This will be performed in the same way as the primary analysis of the primary outcome. We will also perform a sensitivity analysis including those lost to follow-up, by using the most recent observation of this variable. |
| Scar/infiltrate size at 1 week, 3 weeks and 3 months by slit lamp examination | The geometric mean of the two principle axes in mm of the scar or infiltrate at 1 week, 3 weeks and 3 months will be used as a secondary outcome variable. The slit-lamp scar size will be compared at each of these time points between treatment arm in the same manner as described above. This will be by linear regression, with treatment arm and baseline infiltrate/scar size as pre-specified covariates. This controls for the baseline infiltrate/scar size. |
| Time to full epithelial healing (slit lamp examination by ophthalmic clinician) | Time of re-epithelialisation will be defined as the midpoint between the last review where an epithelial defect (ED) was present and the subsequent review where there was no ED. An area of fluorescein staining of less than 0.5 mm will be considered as a resolved ED due to the difficulty in differentiating a smaller defect from a small amount of fluorescein pooling observed in a healed defect.

Analysis of time to healing will use Cox proportional hazards regression with treatment group as the primary predictor and with predictors of baseline ED size (using the geometric mean in mm as outlined above). Survival curves will be plotted using Kaplan-Meier analysis for both treatment arms up to the final visit at 3 months. The proportional hazards assumption will be checked by stratifying on quartiles of the baseline ED size and if the assumption does not hold, the stratified results will be the ones reported. Additionally, treatment failure (defined as persisting epithelial defect greater than 0.5 mm at the 3 month review) will be compared between treatment groups using Fisher's exact test. |
| Rate of healing | We will assess how quickly the area of ulceration reduces over time. The rate will be calculated between the 1-week, 3-week and 3-month review by taking the difference in ED size between the two time points, in mm, and diving by the number of days to give a rate of mm/day. Analysis will be performed using Cox regression. |
| Microbiological cure | Patients who have a persisting corneal ulcer (as defined by the presence of an ED) at day 7 will undergo a repeat corneal scrape and microbiological investigations. Microbiological cure at 7 days will be defined as the absence of any micro-organisms as no significant growth on culture. The number of patients with microbiological cure at day 7 will be compared between the two treatment arms using logistic regression with treatment group and organism (*Aspergillus* spp, *Fusarium* spp, or other) as covariates. |
| Ulcer depth at 1 week and 3 weeks (slit lamp examination by ophthalmic clinician) | The depth of ulcer in terms of percentage of healthy cornea will be compared at 1 week and 3 weeks between treatment arms, adjusting for baseline depth in the same manner with analysis performed by linear regression |
| Hypopyon height at 1 and 3 weeks, (slit lamp examination by ophthalmic clinician) | The hypopyon height in mm will be compared at 1 week and 3 weeks between treatment arms, adjusting for baseline hypopyon height in the same manner with analysis performed in the same way (linear regression) |
| Perforation and/or TPK and/or conjunctival advancement by 3 months (slit lamp examination by ophthalmic clinician) | The number of patients who undergo perforation and/or require TPK and/or have undergone conjunctival advancement by 3 months will be reported using CIs and descriptive statistics. The study is not powered to detect a difference in perforation rate or TPK between treatment groups; not reporting a significant difference may be wrongly interpreted as there being no difference between groups. We will therefore perform an exploratory analysis to compare TPK or perforation rates between treatment groups. This will be by logistic regression to compute an OR by arm. |
| Loss of eye | The number of patients who have their eye surgically removed (evisceration or enucleation) during the 3 months follow-up period will be reported using CIs and descriptive statistics in the same way as for TPK/perforation rate above, along with exploratory analysis using logistic regression to find risk factors for eye loss and OR for this by arm. |

Continued

**Table 4** Continued

| Secondary outcome measure | Details |
|---|---|
| Ocular adverse effects, slit lamp examination by ophthalmic clinician | The proportion of patients with one or more adverse events will be compared using Fisher's exact test. Additional analysis to compare the rate of adverse events during the 3 months follow-up will be by Poisson regression as this can take into account multiple instances within one participant. |
| QoL assessed using: EQ-5D, WHO/PBD-VF20, WHOQOL-BREF | QoL can be assessed quantitatively using different tools depending on what is of interest. For example, disease-related QoL can be assessed (eg, vision related QoL, VRQoL) or more general health-related issues irrespective of the disease can be investigated (health-related QoL, HRQoL).[40] |
| | We will use the WHO/PBD-VF20 (WHO/ Prevention of Blindness and Deafness—Visual Functioning 20-item questionnaire) VRQoL tool. This tool measures the impact of visual impairment in the person's life including mental well-being, dependency and social functioning. These have been used in a number of other visual related studies to show a difference in QoL.[36 37] |
| | For HRQoL, we will use the EQ-5D questionnaire, EQ-Visual Analogue Scale and the WHOQOL-BREF. The EQ-5D is a standardised tool to measure health outcomes.[38] The WHOQOL-BREF has good applicability in (LMIC as it was developed simultaneously from concept across 18 countries in Africa, Asia and Latin America. It measures four domains of health: Physical Health, Psychological Health, Social Relationships, and Environment. Details of this scoring are given in table 2.[39] |
| | Analysis will be by comparing the scores obtained for each QoL assessment for the two treatment arms to estimate the effect of CHX and NATA on patients' QoL. This will be similar to that performed by Habtamu et al.[40] Comparisons between the two medication groups will be adjusted for the matching variables: age and sex. The VRQoL analysis was also adjusted for socio-economic status and the HRQoL analysis adjusted for both socioeconomic status and presence of health problems during the previous 4 weeks, as these factors may confound the association between fungal keratitis and QoL. Logistic, linear and ordinal logistic regression methods will be used for binary, continuous and ordered categorical outcome variable analysis, respectively. Linear regression models and the t-test were employed to compare significant differences in QoL scores and to generate mean and mean differences between the two treatment arms in each QoL subscale and domain, respectively. |
| Cost-effectiveness analysis, using EQ-5D data from 3 months and direct cost data | Direct cost data will be collected at the 3 months follow-up. Economic costs to the patient can also be calculated from the EQ-5D questionnaire, which will be asked at baseline and at the 3 months follow-up. Mean direct costs incurred by patients will be compared between interventional arms using the t-test for significance. The difference from the baseline EQ-5D and the 3 months EQ-5D mean scores will also be compared in a similar fashion. |
| Drug adherence | The rate of drug adherence will be compared between the two treatment groups using descriptive statistics. |

BSCVA, best spectacle corrected visual acuity; EQ-5D, EuroQol-5 dimensions; LMIC, low-income and middle-income countries; QoL, quality of life; TPK, therapeutic penetrating keratoplasty.

significant difference between CHX and NATA, with a Cochrane review finding a non-significant trend favouring CHX over NATA.[10 18 19] Despite this trend favouring CHX, we have chosen to carry out a non-inferiority trial rather than a superiority trial as this is a more clinically pragmatic approach. It would be counterproductive to conduct a superiority trial and find no statistically significant difference between the two treatments leading clinicians to potentially disregard CHX as a treatment, when in fact it may be 'non-inferior' to NATA, and be the only available or most cost-effective treatment.

The choice of delta: This clinically meaningful difference of 0.15 logMAR was chosen as a difference of 0.15 logMAR corresponds to approximately 1.5 lines on a Snellen chart; any difference greater than this is clinically significant, as a difference of less than 0.15 log MAR could potentially be accounted for by testing/retesting error.[29] Furthermore, it was used in the MUTT1 trial, providing methodological consistency between studies.[6] In addition, previous studies have suggested treated ulcers improve at a mean of four Snellen lines from baseline.

Sample size was calculated for 90% power and adjusted final alpha of 0.0492, taking account of a single interim analysis using O'Brien-Fleming approach to maintain type-1 error rate of 5%. A sample size of 452 is required to detect a non-inferiority margin of 0.15-logMAR in BSCVA 3 months after enrolment between arms, assuming 0.5 SD for 3 months BSCVA and 15% drop-out. However, given approximately 10% of infections are mixed and these will be excluded from the primary analysis, 500 patients will be recruited. This sample size provides 90% power to detect superiority in BSCVA at 3 months if there is ≥0.17 LogMAR units difference as a secondary analysis.

### Analysis plan

The analysis will be by intention to treat (ITT). All patient data will be analysed according to their randomisation

**Table 5** Registration data and protocol summary

| Data category | Information |
| --- | --- |
| Primary registry and trial identifying no | ISRCTN Registry; ISRCTN14332621 |
| Date of registration in primary registry | 15 May 2019 |
| Secondary identifying numbers | |
| Source(s) of monetary or material support | Wellcome Trust |
| Primary sponsor | London School of Hygiene and Tropical Medicine |
| Secondary sponsor(s) | |
| Contact for queries | Jeremy Hoffman FRCOphth (Jeremy.hoffman@lshtm.ac.uk) |
| Title | Chlorhexidine 0.2% vs Natamycin 5% for the treatment of fungal corneal infections |
| Countries of recruitment | Nepal |
| Health condition(s) or problem(s) studied | Fungal keratitis |
| Intervention(s) | Participants will be randomised to either topical chlorhexidine 0.2% or topical natamycin 5% |
| Key eligibility criteria | 1. Acute MK characterised by:<br>► Corneal epithelial ulceration >1 mm diameter<br>► Corneal stromal infiltrate<br>► Acute inflammation: for example, conjunctival injection, anterior chamber inflammatory cells, hypopyon<br>2. Filamentous fungal hyphae visualised on smear microscopy and/or in vivo confocal microscopy<br>3. Agree to be randomised to either treatment arm and able to give informed consent<br>4. Agree to be followed up at 2 days, 1 week, 2 weeks, 3 weeks, 2 months and 3 months<br>5. Adults (18 years and older) |
| Study type | Randomised controlled trial |
| Date of first enrolment | 1 June 2019 |
| Target sample size | 500 |
| Recruitment status | Recruiting |
| Primary outcome(s) | Best Spectacle Corrected Visual Acuity at 3 months by a trial certified optometrist |
| Key secondary outcomes | 1. Time to full epithelial healing (slit lamp examination by ophthalmic clinician)<br>2. Pin-hole visual acuity in logMAR at 3 months, trial-certified optometrist<br>3. Scar/infiltrate size at 1 week, 3 weeks and 3 months (slit-lamp examination by ophthalmic clinician)<br>4. Ulcer depth at 1 week and 3 weeks (slit-lamp examination by ophthalmic clinician).<br>5. Hypopyon height at 1 and 3 weeks, (slit-lamp examination by ophthalmic clinician).<br>6. Perforation and/or TPK by 3 months (slit-lamp examination by ophthalmic clinician).<br>7. Positive culture rate at 1 week<br>8. Ocular adverse effects at each follow-up visit (day 2, day 7, day 14, 3 weeks, 2 months, 3 months), slit-lamp examination by ophthalmic clinician<br>9. Quality of life (QoL) assessed using: EQ-5D, WHO/PBD-VF20, WHOQOL-BREF (comparison between baseline and QoL measures at 3 months)<br>10. Cost-effectiveness analysis, using EQ-5D data from 3 months and direct cost data<br>11. Drug adherence at each follow-up visit (day 2, day 7, day 14, 3 weeks, 2 months, 3 months) while the patient is using study medications |

EQ-5D, EuroQol-5 Dimension; LMIC, low-income and middle-income countries; MK, microbial keratitis; TPK, therapeutic penetrating keratoplasty.

allocation irrespective of whether or not the patient received or adhered to the allocated treatment. Consolidated Standards of Reporting Trials guidelines for analysing/reporting non-inferiority RCTs will be followed. A flow chart showing cases assessed, recruited and followed up by arm will be prepared.[32] Baseline characteristics will be summarised by arm. The Standard Protocol Items: Recommendations for Interventional Trials checklist is given in online supplementary appendix 4.

## Primary outcome analysis: unadjusted analysis

The primary analysis will be conducted using all available data, missing data due to lost to follow-up will be excluded. The primary analysis of the primary outcome (BSCVA at 3 months) will be by linear regression, with treatment arm and baseline BSCVA as prespecified covariates. This controls for the baseline BSCVA. The treatment group is the primary predictor. Primary analysis will exclude mixed fungal and bacterial infection (isolated in the baseline sample).

We will use our alpha of 0.0492 to test the null hypotheses at 0.0492 significance. The null hypothesis for non-inferiority is that the mean BSCVA at 3 months for CHX is greater than or equal to 0.15 logMAR worse than the BSCVA when natamycin is used. Significance will be assessed using a two-tailed test at 0.0492 level for assessing non-inferiority. CHX will be non-inferior if the upper one-sided 95% confidence level for this regression coefficient (ie, the effect of CHX controlling for baseline BSCVA) exceeds 0.15 logMAR.

## Primary outcome analysis: adjusted analysis

In the event that there is a baseline imbalance between the treatment groups in a baseline covariate due to chance, we will perform an adjusted (sensitivity) analysis (see below). This is particularly important if CHX has a better outcome than NATA, as the adjusted treatment effects may account for this observed imbalance while the unadjusted analyses may not. Sensitivity analyses will allow us to show that any observed positive treatment effect is not solely explained by imbalances at baseline in any of the covariates.

## Secondary analyses of the primary outcome
### Per-protocol analysis

Repeat analysis of the primary outcome will be done as per the protocol, based on what the participants actually took. The per-protocol population will include all the individuals included in the primary ITT analysis, excluding individuals who showed poor compliance with the medications (defined as taking less than 50% through self-reporting or bottle weighing, whichever is the lower); individuals where there have been major protocol deviations; and non-fungal or mixed corneal infections (should these patients happened to have been randomised). All analyses that are performed as ITT will be repeated as per protocol and labelled as such.

### Mixed infections

Secondary analysis of the primary outcome (by ITT) will include mixed infections and will be carried out in the same way as in primary analysis of the primary outcome above.

### Sensitivity analyses

For the primary analysis those individuals with missing outcome data (ie, lost to follow-up) will be excluded. Sensitivity analyses will be performed by imputing a range of scenarios to demonstrate a range of potential results, where there is missing outcome data. In the case of substantial missing data in the trial, the primary analysis will be carried out as previously stated excluding missing observations. This, however, assumes data are missing completely at random. As a sensitivity analysis, we will then apply a multiple imputation approach to the missing data, if we consider the data are randomly missing conditional on other observed covariates.

In the case that there is a systematic (ie, non-random) reason for a difference in the follow-up rates between the two groups, we will explore models in which these missing outcome data are assumed to be non-random, that is, dependent on the outcome being regressed, BSCVA or treatment group.

We will also perform sensitivity analyses on patients whose vision is CF or worse, those patients who have undergone a TPK or those with corneal perforation, to see if there is any change in the effect size or conclusions drawn.

### Analysis of other potential determinants for success

Logistic regression random-effects models will be used to analyse potential factors that may be associated with a poor primary outcome, BSCVA at 3 months, defined as >1.0 logMAR. Individual baseline characteristics will be used separately as an exposure variable with BSCVA at 3 months as the outcome, with the model adjusted for trial arm. A multivariate model will be built using parameters with a p<0.2 in the log likelihood ratio test. Variables will be removed one by one, by omitting the variable with the largest p value each time, until all predictors in the model have a p<0.05.

## Secondary outcome analysis

The secondary outcomes outlined above and detailed in table 4 will be analysed by arm. Additional adjustment for factors imbalanced between arms at baseline will be introduced as appropriate. Continuous outcomes will be analysed using linear regression. Binary and ordinal outcomes will analysed using logistic regression. Details of these are given in table 4.

## Interim analysis

An interim analysis will be conducted for the DSMB by an independent statistician after 1/3 of patients recruited have completed follow-up.

## Patient and public involvement

Mixed methods descriptive cross-sectional studies with semistructured interviews and focus group discussions with patients and eye health providers were carried out in Sagarmatha zone, Eastern Nepal at various points in 2018.[33] These conversations highlighted the delayed presentation often seen with FK combined with the often prohibitively high costs of treatment. Furthermore, treatment is not always felt to be effective. Eye health workers were keen to receive further training and highlighted the need for greater government support in the provision of eye care services in the community.

## ETHICS AND DISSEMINATION

Ethics committee and regulatory review and approval has been obtained from the Nepal Health Research Council (NHRC) Ethics Committee, Kathmandu, Nepal; the Department of Drug Administration (DDA), Kathmandu, Nepal; and the London School of Hygiene and Tropical Medicine Ethics Committee, UK. The trial is registered with the ISRCTN clinical trials registry. Protocol modifications are submitted to the relevant parties for review and/or approval. Table 5 summarises the study protocol and trial registration information. At the end of the study period, patients who still require treatment or follow-up will continue to be treated at SCEH as per routine clinical practice. The trial sponsor is the LSHTM. The results of this trial will be presented at local and international meetings and submitted to peer-reviewed journals for publication.

### Author affiliations
[1]International Centre for Eye Health, London School of Hygiene and Tropical Medicine, London, UK
[2]Cornea Department, Sagarmatha Choudhary Eye Hospital, Lahan, Nepal
[3]Eastern Region Eye Care Programme, Biratnagar, Nepal
[4]Mbarara University of Science and Technology Faculty of Medicine, Mbarara, Uganda
[5]Department of Ophthalmology, Kilimanjaro Christian Medical Centre, Moshi, Tanzania
[6]MRC Tropical Epidemiology Group, London School of Hygiene & Tropical Medicine, London, UK
[7]Department of External Eye Disease, Moorfields Eye Hospital NHS Foundation Trust, London, UK

**Acknowledgements** The authors would like to thank the Eastern Region Eye Care Programme (EREC-P), Nepal Netra Jyoti Sangh (NNJS) and the Nepal Health Research Council (NHRC) for helping with study coordination and implementation. The authors would like to thank the staff and management board at Sagarmatha Choudhary Eye Hospital (SCEH) for their continued support, coordination and implementation of the study. The authors are grateful to the guidance of the Data Safety Monitoring Board including Reeta Gurung (chair), Salma KC Rai, Sabina Shrestha and Meenu Chaudhary.

**Collaborators** SCEH: Abhishek Roshan (Hospital Manager); Sanjay Kumar Singh (Medical Superintendent); Reena Yadav (Primary Investigator); Sandip Das Sanyam (Study Co-Ordinator); Pankaj Chaudhary (Microbiologist); Rabi Shankar Sah, Kamlesh Yadav (Investigators); Ram Narayan Bhandari, Aasha Chaudhary, Sharban Mandal (Eye Health Workers); Raja Ram Mahato (Randomisation Administrator and Logistics); Lalita Rajbanshi (Laboratory Assistant); Ramesh Sah, Arvind Ray, Sachindra Kamti (Optometrists); Avinash Chaudhary (Ophthalmic Assistant); Padma Narayan Chaudhary (Hospital Chairman); Suresh Singh, Ravi Pant, Rakesh Singh (Hospital Management); Ram Kumar Jha (Ophthalmic Assistant, Rajbiraj ECC). NNJS: Sailesh Kumar Mishra (Executive Director); Sabita KC (Board Secretary); Ranjan Shah (Programme Associate); Jaganath Dhital (Assistant). EREC-P: Sanjay Kumar Singh (Programme Director). JEH: Hemchandra Jha (Medical Superintendent); Mahesh Yadav (Investigator); Rudal Prasad Sah (Ophthalmic Assistant). LSHTM: Jeremy Hoffman (Primary Investigator); Matthew Burton (Chief Investigator); Astrid Leck (Microbiologist); David Macleod, Helen Weiss (Statisticians); Victor Hu (Investigator); Sarah O'Regan (Administrator).

**Contributors** Searched the literature: JJH and MJB. Drafted initial protocol: JJH. Contributed to protocol development and revision: JJH, RY, SDS, PC, AR, SKS, SA, EM, DM, HAW, AL, VH and MJB. Drafted this manuscript: JJH. Critically revised this manuscript: JJH, RY, SDS, PC, AR, SKS, SA, EM, DM, HAW, AL, VH and MJB. Conceptualisation: MJB. Funding acquisition: MJB.

**Funding** This research was funded through a Senior Research Fellowship to MJB from the Wellcome Trust (207472/Z/17/Z).

**Disclaimer** The funders had no role in study design, data collection and analysis, decision to publish, or preparation of the manuscript.

**Competing interests** None declared.

**Patient consent for publication** Obtained.

**Provenance and peer review** Not commissioned; externally peer reviewed.

**ORCID iDs**
Jeremy John Hoffman http://orcid.org/0000-0001-9454-2131
Sandip Das Sanyam http://orcid.org/0000-0002-0554-8441
Helen Anne Weiss http://orcid.org/0000-0003-3547-7936

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
