## [Reviewer comments · BMJ Open]

ARTICLE DETAILS

TITLE (PROVISIONAL)	Topical chlorhexidine 0.2% versus topical natamycin 5% for fungal keratitis in Nepal: rationale and design of a randomised controlled non-inferiority trial
AUTHORS	Hoffman, Jeremy; Yadav, Reena; Das Sanyam, Sandip; Chaudhary, Pankaj; Roshan, Abhishek; Singh, Sanjay; Arunga, Simon; Matayan, Einoti; Macleod, David; Weiss, Helen; Leck, Astrid; Hu, Victor; Burton, Matthew J

VERSION 1 – REVIEW

REVIEWER	Dr Pranita Sahay Lady Hardinge Medical College, New Delhi, India
REVIEW RETURNED	14-Apr-2020

GENERAL COMMENTS	The authors have done a commendable job by conducting a well-designed study comparing the standard antifungal agent (natamycin) with chlorhexidine about which literature is limited and results are inconclusive. The study methodology is indeed good. There are few minor concerns. 1. the term MK/FK are being used interchangeably. Please use one term. also the abbreviation MK has not been spelled out at its first mention.2. were cases of polymicrobial keratitis (with bacterial keratitis) excluded? If so please add this to the exclusion criteria in stage 2. as gram stain would have given a clue for the same at this stage3. why is the term g-chlorhexidine and g-natamycin used in abstract and not elsewhere. i suggest uniformity to be maintained.4. one of the major concern is keeping visual acuity as the primary outcome for this study. The goal of management of all cases of infectious keratitis is rapid infection control and healing of the ulcer. Visual acuity gain is never the primary agenda of management of cases with infectious keratitis, though it can be a secondary outcome measure. Hence, it would have been better if the authors would have kept "time to healing" and "culture negativity at day 7" as the primary outcome measure.
--

REVIEWER	Dr Catherine Fullwood Manchester University NHS Foundation Trust UK
REVIEW RETURNED	15-Apr-2020

GENERAL COMMENTS	Thanks for the opportunity to review your protocol, in general I found it interesting and well written. Below I highlight the odd couple of errors and a few places where I feel some extra clarity is required. Page 5, Article Summary, Point 5: Please add a full stop at the end. Page 10, In vivo confocal microscopy: Please add "is" to the phrase
--

	“the objective lens and is changed between...” Page 11, HIV testing: Please change “and” to “an” in the phrase “unknown to the HIV care services and appropriate referral...” Page 11, Quality of life: Please add details on the scoring for WHOQL-BREF Page 15, Secondary outcome measures: I find the wording of Table 3 a bit confusing. There are no previous details regarding the analysis of the primary outcome, and neither would I expect to see it in this section. However Table 3 seems to imply that prior information is given – where would I find it? Personally I’d rather see a table which describes the outcomes with their time-points and measurement details, leaving the analysis to the analysis section. Having said that the table is clear and would perhaps serve better being moved later on in the protocol. Page 16, line 7: Where is the 6.3 referred to here? Page 16, Perforation and/or TPK and/or conjunctival advancement by three months (slit lamp examination by ophthalmic clinician): This is a bit messy at the end where the exploratory analysis is discussed. Are you mainly looking at TPK / perforation rate or assessing risk factors? Please reword the last 3 sentences. Page 16, quality of life: Do you plan to adjust the linear regression – this sentence almost seems incomplete or as an afterthought compared with the details above. Also do you have details for the scoring of the questionnaires – any references? This relates to the point from page 11. Cost effectiveness – how and when will the costs be collected, what will you look at? Page 11, line 49: this should read “from” not “form” How will adherence be collected / measured? I see that you will measure and compare the amount of medication left with that expected, but what will the actual outcome measure be? Is it a ratio? Do you also have yes/no type measures from patient report? Page 19/20, lines 60-3: Please reword the sentence “CONSORT guidelines for analysing/reporting non-inferiority RCTs will be followed including a flowchart showing cases assessed, recruited and followed-up by arm will be prepared.” Page 19, line 22: Please change “CHX would be found non-inferior..” to read “CHX will be” Page 20, line 47: Please change “per-protocol population would include...” to read “per-protocol population will include Page 21 Secondary outcome analysis: As suggested above some of the details of Table 3 are more appropriate here and make more sense. Page 21, line 60: Please change “of patients recruited and completed follow-up.” to “of patients recruited have completed follow-up.
--	--

REVIEWER	Darren S J Ting
----------	-----------------

	Academic Ophthalmology, University of Nottingham, UK.
REVIEW RETURNED	15-Apr-2020

GENERAL COMMENTS	Fungal keratitis is a common cause of corneal blindness globally, particularly in developing countries. The authors have submitted a very well-designed RCT protocol with an aim to examine the effectiveness and safety of topical natamycin versus chlorhexidine. I only have minor comments for this manuscript. Abstract 1. As this manuscript is for general readership, please consider changing g- to topical (e.g. g-natamycin to topical natamycin). Introduction A good rationale was provided in the introduction section. I wonder if there is any background information on the profile of fungal organisms for Nepal (e.g. filamentous vs. yeast) since natamycin works best for filamentous fungal keratitis as highlighted by the authors? Presumably yeast fungal keratitis is excluded from this study? Methods and analysis Trial summary Will the authors recruit the patients if the patients are “positive” on IVCM but negative on smear? If so, it would be useful to know the specificity of IVCM. And will they only start the randomised treatment after the results become available (but that may delay the treatment)? The authors have provided a comprehensive list of inclusion criteria for the study. This could be potentially further improved if the authors could also consider the following: 1. Will mixed infection (e.g. bacteria + fungi) be excluded from the study? 2. Will they include or exclude non-filamentous fungal keratitis (e.g. candida keratitis)? 3. Will they also include cases presenting with threatened / actual corneal perforation? As BSCVA (the primary outcome measure) is known to be affected by the location of the ulcer and severity of ulcer at initial presentation, will there be any plan to grade the severity of ulcer at recruitment and ensure that similar or equal portion of patients with similar severity randomised to each treatment arm?
--

REVIEWER	Mohammad Soleimani Farabi Eye Hospital, Tehran University of Medical Sciences, Tehran, Iran
REVIEW RETURNED	16-Apr-2020

GENERAL COMMENTS	Many thanks for your collaborative study, fungal keratitis is a very important ocular problem in developing countries. I agree that there is a limited resource for fungal keratitis; however, I have great concern regarding the role of chlorhexidine 0.2% in treating patients with mycotic keratitis. To me, it is difficult to justify the ethical issue and rely on a few studies regarding the effect of chlorhexidine vs. documented studies for natamycin. 2. In many of these patients with fungal keratitis, it is difficult to find a reliable BCVA in consecutive follow-ups especially in patients with central keratitis. So it may be better to assess the time to re-epithelialization for the primary outcome.
--

VERSION 1 – AUTHOR RESPONSE

Reviewer 1

The authors have done a commendable job by conducting a well-designed study comparing the standard antifungal agent (natamycin) with chlorhexidine about which literature is limited and results are inconclusive. The study methodology is indeed good. There are few minor concerns.

1. the term MK/FK are being used interchangeably. Please use one term. Also the abbreviation MK has not been spelled out at its first mention.

Thank you for this feedback. We have corrected this in the manuscript and defined the abbreviation MK at its first usage. We have also changed “MK” to “FK” where appropriate to ensure that there is no confusion. However, in some instances we have left the term MK as this specifically refers to patients who, on first presentation to the hospital, have an unknown causative organism. It is important to distinguish these MK patients to those with proven fungal pathogens (FK). We have therefore left the term MK in a few specific cases in the methodology, including in the eligibility criteria.

2. were cases of polymicrobial keratitis (with bacterial keratitis) excluded? If so please add this to the exclusion criteria in stage 2. as gram stain would have given a clue for the same at this stage

Mixed infections were included in recruitment but will be excluded from the primary analysis. Although this is mentioned within the analysis, we acknowledge that it is not explicitly mentioned. We have therefore added the following text to the end of the eligibility section (Page 7):

As some patients will be enrolled on the basis of the results of *in vivo* confocal microscopy which is unable to detect most bacteria reliably, some patients with microscopically confirmed fungal infection will subsequently also be found to have had mixed infection at the time of being recruited into the study, as bacterial cultures may become positive a few days after enrolment. Based on previous experience at SCEH this is likely to account for about 10% of cases. These patients are included in the study but excluded from the primary analysis of the primary outcome (see below). Secondary analyses will include mixed infections.

3. why is the term g-chlorhexidine and g-natamycin used in abstract and not elsewhere. I suggest uniformity to be maintained.

Thank you for this comment. We have clarified this by removing the “g-“ and replacing this with “topical”. This will also make this easier to understand for non-ophthalmologists.

4. one of the major concerns is keeping visual acuity as the primary outcome for this study. The goal of management of all cases of infectious keratitis is rapid infection control and healing of the ulcer. Visual acuity gain is never the primary agenda of management of cases

with infectious keratitis, though it can be a secondary outcome measure. Hence, it would have been better if the authors would have kept "time to healing" and "culture negativity at day 7" as the primary outcome measure

We appreciate this comment from the reviewer and it is something that we considered in depth prior when designing the study and writing the protocol. The justification for using best spectacle corrected visual acuity (BSCVA) as the primary outcome measure is given in the Outcome Measures section. We chose BSCVA as this has been the primary outcome of several other trials of the treatment of FK and will therefore facilitate comparison. Three months is the time at which clinical experience suggests most corneal ulcers have usually

healed. BSCVA is chosen as it is a robust measure and is of functional significance. We understand that improvement in visual acuity is not the primary agenda for the treating clinician in the initial weeks of managing such cases – the first step is to save the eye, then focus on visual rehabilitation. For this reason, we already have time to healing and culture negativity at day 7 as secondary outcome measures.

Reviewer 2

Thanks for the opportunity to review your protocol, in general I found it interesting and well written. Below I highlight the odd couple of errors and a few places where I feel some extra clarity is required.

Page 5, Article Summary, Point 5: Please add a full stop at the end.

Thank you for this comment. We have corrected this.

Page 10, In vivo confocal microscopy: Please add “is” to the phrase “the objective lens and is changed between...”

Thank you for this comment. We have corrected this.

Page 11, HIV testing: Please change “and” to “an” in the phrase “unknown to the HIV care services and appropriate referral...”

Thank you for this comment. We have corrected this.

Page 11, Quality of life: Please add details on the scoring for WHOQL-BREF

We have clarified this section and added details on the scoring for EQ-5D with a reference. WHOQoL-BREF was described in the table but it was not originally clear if this was what was being described. This has now been clarified.

Page 15, Secondary outcome measures: I find the wording of Table 3 a bit confusing. There are no previous details regarding the analysis of the primary outcome, and neither would I expect to see it in this section. However Table 3 seems to imply that prior information is given – where would I find it? Personally I’d rather see a table which describes the outcomes with their time-points and measurement details, leaving the analysis to the analysis section. Having said that the table is clear and would perhaps serve better being moved later on in the protocol.

Thank you for this comment. We have moved the Table so that it now appears within the Analysis Plan section of the manuscript as we agree that this makes more sense.

Page 16, line 7: Where is the 6.3 referred to here?

Thank you for this comment. We have corrected this and removed the reference to 6.3 as this was used in an earlier draft of the manuscript prior to submission.

Page 16, Perforation and/or TPK and/or conjunctival advancement by three months (slit lamp examination by ophthalmic clinician): This is a bit messy at the end where the exploratory analysis is discussed. Are you mainly looking at TPK / perforation rate or assessing risk factors? Please reword the last 3 sentences.

We have re-worded this section so that it now reads (Page 19):

The study is not powered to detect a difference in perforation rate or TPK between treatment groups; not reporting a significant difference may be wrongly interpreted as there being no difference between groups. We will therefore perform an exploratory analysis to compare TPK or perforation rates between treatment groups. This will be by logistic regression to compute an odds ratio by arm.

Page 16, quality of life: Do you plan to adjust the linear regression – this sentence almost seems incomplete or as an afterthought compared with the details above. Also do you have details for the scoring of the questionnaires – any references? This relates to the point from page 11.

We have added more detail to this section to explain how we intend to analyse quality of life.

This section now reads (Page 19):

Analysis will be by comparing the scores obtained for each QoL assessment for the two treatment arms to estimate the effect of CHX and NATA on patients' quality of life. This will be similar to that performed by Habtamu et al. Comparisons between the two medication groups will be adjusted for the matching variables: age and sex. The VRQoL analysis was also adjusted for socio-economic status and the HRQoL analysis adjusted for both socio-economic status and presence of health problems during the previous four weeks, as these factors may confound the association between microbial keratitis and QoL. Logistic, linear and ordinal logistic regression methods will be used for binary, continuous and ordered categorical outcome variable analysis, respectively. Linear regression models and the t-test were employed to compare significant differences in QoL scores and to generate mean and mean differences between the two treatment arms in each QoL subscale and domain, respectively.

Cost effectiveness – how and when will the costs be collected, what will you look at?

We have added further information to this section now so that there are details on how and when cost effectiveness data will be collected. This section now reads (Page 19):

Direct cost data will be collected at the three-month follow-up. Economic costs to the patient can also be calculated from the EQ-5D questionnaire, which will be asked at baseline and at the three-month follow-up. Mean direct costs incurred by patients will be compared between interventional arms using the t-test for significance. The difference from the baseline EQ-5D and the three-month EQ-5D mean scores will also be compared in a similar fashion.

Page 11, line 49: this should read “from” not “form”

Thank you for this comment. We have corrected this.

How will adherence be collected / measured? I see that you will measure and compare the amount of medication left with that expected, but what will the actual outcome measure be? Is it a ratio? Do you also have yes/no type measures from patient report?

We have added more information here (Page 13) to describe that we will also be collecting information on adherence using the Case Record File at each visit. The measure for the amount of

medication left to expected will be presented as a ratio and compared between groups using descriptive statistics. This has been added to the manuscript.

Page 19/20, lines 60-3: Please reword the sentence “CONSORT guidelines for analysing/reporting non-inferiority RCTs will be followed including a flowchart showing cases assessed, recruited and followed-up by arm will be prepared.”

Thank you for this comment. We have reworded this.

Page 19, line 22: Please change “CHX would be found non-inferior..” to read “CHX will

be”

Thank you for this comment. We have reworded this.

Page 20, line 47: Please change “per-protocol population would include...” to read “per-protocol population will include

Thank you for this comment. We have reworded this.

Page 21 Secondary outcome analysis: As suggested above some of the details of Table 3 are more appropriate here and make more sense.

Thank you for this comment. We have moved the table to this section as it is more appropriate here.

Page 21, line 60: Please change “of patients recruited and completed follow-up.” to “of patients recruited have completed follow-up

Thank you for this comment. We have reworded this.

Reviewer 3

Fungal keratitis is a common cause of corneal blindness globally, particularly in developing countries. The authors have submitted a very well-designed RCT protocol with an aim to examine the effectiveness and safety of topical natamycin versus chlorhexidine. I only have minor comments for this manuscript.

Abstract

1. As this manuscript is for general readership, please consider changing g- to topical (e.g. g-natamycin to topical natamycin).

Thank you for this comment. We have corrected this.

Introduction

A good rationale was provided in the introduction section. I wonder if there is any background information on the profile of fungal organisms for Nepal (e.g. filamentous vs. yeast) since natamycin works best for filamentous fungal keratitis as highlighted by the authors? Presumably yeast fungal keratitis is excluded from this study?

Thank you for this comment. The vast majority of cases of fungal keratitis in Nepal are caused by filamentous fungi with yeast infections only accounting for a small minority of cases. As the reviewer

has quite rightly stated, natamycin is not an effective treatment for yeast infections. Only cases of filamentary fungal keratitis are included or mixed bacterial and filamentary fungal cases.

Methods and analysis

Trial summary

Will the authors recruit the patients if the patients are “positive” on IVCM but negative on smear? If so, it would be useful to know the specificity of IVCM. And will they only start the randomised treatment after the results become available (but that may delay the treatment)?

Patients are eligible for the study if they have fungal filaments visible IVCM and / or if there are fungal hyphae visible on light microscopy. There is no delay to commencing treatment as the results of IVCM are available immediately as it is a “real-time” process. Results for light microscopy are available within 15 minutes of the corneal scrape being taken in our current setup.

There is strong evidence supporting the use of IVCM for diagnosing filamentary fungal keratitis. Studies have reported specificities of between 81.4 and 92.7%.^{1,2}

The authors have provided a comprehensive list of inclusion criteria for the study. This could be potentially further improved if the authors could also consider the following:

- 1. Will mixed infection (e.g. bacteria + fungi) be excluded from the study?**
- 2. Will they include or exclude non-filamentous fungal keratitis (e.g. candida keratitis)?**
- 3. Will they also include cases presenting with threatened / actual corneal perforation?**

1. Mixed infections are included in recruitment but will be excluded from the primary analysis. Although this is mentioned within the analysis, we acknowledge that it is not explicitly mentioned. We have therefore added the following text to the end of the eligibility section (Page 7):

As some patients will be enrolled on the basis of the results of *in vivo* confocal microscopy which is unable to detect most bacteria reliably, some patients with microscopically confirmed fungal infection will subsequently also be found to have had mixed infection at the time of being recruited into the study, as bacterial cultures may become positive a few days after enrolment. Based on previous experience at SCEH this is likely to account for about 10% of cases. These patients are included in the study but excluded from the primary analysis of the primary outcome (see below). Secondary analyses will include mixed infections.

2. Only filamentary fungi are eligible for recruitment to the study.

3. Cases that have already perforated or will imminently perforate are deemed “Very severe ulcers warranting immediate evisceration or conjunctival flap” and are excluded from recruitment. Such cases are unlikely to respond to any form of medical management and will require surgical intervention.

As BSCVA (the primary outcome measure) is known to be affected by the location of the ulcer and severity of ulcer at initial presentation, will there be any plan to grade the severity of ulcer at recruitment and ensure that similar or equal portion of patients with similar severity randomised to each treatment arm?

This is an important consideration as both the location of the ulcer and the severity do affect the presenting BSCVA as the reviewer has quite rightly stated. All recruited patients have the epithelial defect and stromal infiltration measured in a standardised way at presentation and over the course of the study at the regular follow-ups. The location of the ulcer is also categorised at baseline and follow-up. Rather than choosing to randomise an equal proportion of ulcers of different severity / location, which poses methodological challenges, patients will simply be randomised in a 1:1 allocation ratio of CHX to NATA, with a random block size (2, 4 or 6). Primary analysis will be unadjusted to the size or location of the presenting corneal ulcer. However, sensitivity analyses will be conducted as detailed in the Data Analysis Plan SOP. This states:

Adjusted analysis

In the event that there is a baseline imbalance between the treatment groups in a baseline covariate due to chance, we will perform an adjusted (sensitivity) analysis (see below). This is particularly important if CHX has a better outcome than NATA, as the adjusted treatment effects may account for this observed imbalance whilst the unadjusted analyses may not. Sensitivity analyses will allow us to show that any observed positive treatment effect is not solely explained by imbalances at baseline in any of the covariates.

Analysis of Other Potential Determinants for Success

Logistic regression random effects models will be used to analyse potential factors that may be associated with a poor primary outcome, BSCVA at 3 months, defined as >1.0 logMAR. Individual baseline characteristics will be used separately as an exposure variable with BSCVA at 3 months as the outcome, with the model adjusted for trial arm. A multivariate model will be built using parameters with a p-value of < 0.2 in the log likelihood ratio test. Variables will be removed one by one, by omitting the variable with the largest p-value each time, until all predictors in the model have a p<0.05.

Characteristics to be considered in the analysis:

- Trial arm
- Socio-demographic factors: o
age

- o sex
- Clinical status / pre-treatment disease severity:
 - o Visual acuity
 - o size of epithelial defect
 - o location of the epithelial defect
 - o presence of hypopyon
- Clinical history:
 - o Time delay to presentation
 - o Prior and concurrent medication o use of steroids
 - o use of TEM
 - o prior use of antifungals o prior use of antibiotics
- species of fungus
- Treatment compliance
- Presence of mixed infection

Reviewer 4

Many thanks for your collaborative study, fungal keratitis is a very important ocular problem in developing countries. I agree that there is a limited resource for fungal keratitis; however, I have great concern regarding the role of chlorhexidine 0.2% in treating patients with mycotic keratitis. To me, it is difficult to justify the ethical issue and rely on a few studies regarding the effect of chlorhexidine vs. documented studies for natamycin.

We strongly disagree with this comment. Chlorhexidine (CHX) is an antiseptic agent, with both antibacterial and antifungal properties. It is a widely-used broad-spectrum biocide, killing microorganisms through cell membrane disruption.³⁻⁵ For example, chlorhexidine 0.2% w/v solution is very widely used as a long-term mouth wash for the prevention and treatment of oral candidiasis (a fungal infection) and for general oral hygiene.⁶⁻⁸ Chlorhexidine 0.2% mouth wash is considered to be locally and systemically safe.

CHX has been used in ophthalmology for more than 30 years as an eye-drop preservative, sterilizing contact lenses, pre-operative topical antiseptic and for treating *Acanthamoeba sp.* and fungal keratitis.⁹⁻¹⁴ It is very important to note that all chlorhexidine solutions used topically in ophthalmic practice are aqueous preparations: i.e. they must not contain any detergents or alcohol.

In a study evaluating potential affordable anti-fungal treatments for keratitis, chlorhexidine digluconate was compared *in vitro* with propamidine (Brolene), povidone iodine and polyhexamethylene biguanide (PHMB).¹⁵ Several concentrations of these agents were tested against a panel of 95 fungal keratitis isolates from Ghana and India. The chlorhexidine 0.2% gave the best results *in vitro*, with inhibition of 90/95 isolates. The investigators then conducted a pilot case series study in India of chlorhexidine digluconate 0.2% in 11 patients with fungal keratitis (7 non-severe and 4 severe cases). They found that 10/11 cases healed on CHX; one severe case did not respond. The study also included a non-randomised comparison group of 8 patients with fungal keratitis (7 non-severe, 1 severe) who were

treated with topical econazole (a frequently used treatment at that time). They reported that 7/8 responded to econazole; the severe case did not respond to the econazole.

Subsequently two pilot RCTs of CHX for fungal keratitis were conducted. In the first trial, involving 60 patients conducted in south India, three chlorhexidine gluconate concentrations (0.05%, 0.1%, 0.2% w/v) were compared to each other and to natamycin 5%.¹¹ There was evidence suggestive that chlorhexidine 0.2% might be better than natamycin 5% both in terms of the proportion showing a favourable response by 5 days (75% vs. 44%) and cure by 21 days (83% vs. 50%). The CHX 0.2% performed better than both the 0.05% and the 0.1% concentrations. The chlorhexidine 0.2% w/v concentration used in this trial is the same as that is used in mouthwash; and is systemically safe for oral mucosal application.

In the second trial, involving 70 patients conducted in Bangladesh, CHX 0.2% was compared to topical NATA 2.5% (half standard concentration). There was evidence CHX 0.2% was associated with a favourable response in more cases than NATA 2.5% by 5 days (89% vs. 51%; RR=0.23, 95%CI 0.09-0.63).¹⁰ By 21 days 44% of the CHX treated group were cured compared to 28% of the NATA group.

Overall, a Cochrane systematic review of treatments for fungal keratitis found a non-significant trend favouring CHX over NATA in “curing” by 21-days (RR=0.70, 95%CI 0.45-1.09), when the data from these two trials was combined.¹⁶

In the first RCT from India, no toxicity effects were observed. In the second RCT from Bangladesh both CHX 0.2% and NATA 2.5% were well-tolerated; no treatment was discontinued because of allergy or toxicity. One patient in the chlorhexidine arm developed

short-lived punctate corneal epithelial changes, which resolved when the drop frequency was reduced. This a common finding when many different antibiotic drops are used very frequently. There was no early cataract development up to one year. Overall, CHX is safe and well-tolerated at these concentrations when used as a topical treatment for corneal infections.⁹⁻¹⁴

¹⁷ There is also extensive experience from using topical CHX for other indications. For example, CHX is applied to the ocular surface for antisepsis before giving intravitreal injections. A recent published case series from Australia of 40,535 intravitreal injections which used CHX 0.1% or 0.05% for antisepsis reported that it was well tolerated, with only one suspected mild local allergic reaction noted.¹⁸

There have been several reports in the dermatology surgery literature of corneal toxicity following the use of skin antisepsis solutions containing chlorhexidine (concentrations 0.5 – 4%) applied to the face.¹⁹⁻²² However, in all of these cases the solutions contained alcohol and detergent, which are known to be harmful to the eyes and were the likely cause of the toxicity. Solutions containing alcohol or detergent must not be used near the eye as these are harmful.

It is important to note that the chlorhexidine 0.2% eye drop solution used in the earlier trials and in clinical practice only contains water and no excipients.

CHX is used for treating fungal MK in the UK, EU, US and SSA.^{9 23 24} We use CHX for this indication, and have seen responses in eyes deteriorating on NATA. In our experience it is usually well-tolerated by patients.

The primary aim of this study is to answer this question definitively in an adequately powered study.

Regarding any ethical concerns, this study has been approved by the following ethics and regulatory committees:

- The London School of Hygiene and Tropical Medicine
- Nepal Health Research Council
- Nepal Department of Drug Administration

A parallel study using the same methodology as described in this manuscript and supporting documents that will be conducted and analysed separately to this study in Nepal, has been approved to start recruitment in East Africa, having undergone rigorous ethical review by the following institutions:

- The London School of Hygiene and Tropical Medicine
- National Institute for Medical Research, Tanzania
- Uganda National Council for Science and Technology
- Mbarara University of Science and Technology
- Kilimanjaro Christian Medical College

- Tanzania Medicine and Medical Devices Authority

1. In many of these patients with fungal keratitis, it is difficult to find a reliable BCVA in consecutive follow-ups especially in patients with central keratitis. So it may be better to assess the time to re-epithelialization for the primary outcome.

We appreciate this comment from the reviewer and it is something that we considered in depth prior when designing the study and writing the protocol. The justification for using best spectacle corrected visual acuity (BSCVA) as the primary outcome measure is given in the Outcome Measures section. We chose BSCVA as this has been the primary outcome of several other trials of the treatment of FK and will therefore facilitate comparison. Three

months is the time at which clinical experience suggests most corneal ulcers have usually healed. BSCVA is chosen as it is a robust measure and is of functional significance. We understand that improvement in visual acuity is not the primary agenda for the treating clinician in the initial weeks of managing such cases – the first step is to save the eye, then focus on visual rehabilitation. For this reason, we already have time to healing and culture negativity at day 7 as secondary outcome measures.

References

1. Chidambaram JD, Prajna NV, Larke NL, et al. Prospective Study of the Diagnostic Accuracy of the In Vivo Laser Scanning Confocal Microscope for Severe Microbial Keratitis. *Ophthalmology* 2016;123(11):2285-93. doi: 10.1016/j.ophtha.2016.07.009
2. Vaddavalli PK, Garg P, Sharma S, et al. Role of confocal microscopy in the diagnosis of fungal and acanthamoeba keratitis. *Ophthalmology* 2011;118(1):29-35. doi: 10.1016/j.ophtha.2010.05.018
3. McDonnell G, Russell AD. Antiseptics and disinfectants: activity, action, and resistance. *Clinical Microbiology Reviews* 1999;12(1):147-79.
4. Shariff JA, Lee KC, Leyton A, et al. Neonatal mortality and topical application of chlorhexidine on umbilical cord stump: a meta-analysis of randomized control trials. *Public health* 2016;139:27-35. doi: 10.1016/j.puhe.2016.05.006
5. Zhou J, Hu B, Liu Y, et al. The efficacy of intra-alveolar 0.2% chlorhexidine gel on alveolar osteitis: a meta-analysis. *Oral diseases* 2017;23(5):598-608. doi: 10.1111/odi.12553
6. James P, Worthington HV, Parnell C, et al. Chlorhexidine mouthrinse as an adjunctive treatment for gingival health. *The Cochrane database of systematic reviews* 2017;3(1):CD008676. doi: 10.1002/14651858.CD008676.pub2
7. Nittayananta W, DeRouen TA, Arirachakaran P, et al. A randomized clinical trial of chlorhexidine in the maintenance of oral candidiasis-free period in HIV infection. *Oral diseases* 2008;14(7):665-70. doi: 10.1111/j.1601-0825.2008.01449.x
8. Ellepola AN, Samaranyake LP. Adjunctive use of chlorhexidine in oral candidoses: a review. *Oral diseases* 2001;7(1):11-17.
9. Ong HS, Fung SSM, Macleod D, et al. Altered Patterns of Fungal Keratitis at a London Ophthalmic Referral Hospital: An Eight-Year Retrospective Observational Study. *American journal of ophthalmology* 2016;168:227-36. doi: 10.1016/j.ajo.2016.05.021
10. Rahman MR, Johnson GJ, Husain R, et al. Randomised trial of 0.2% chlorhexidine gluconate and 2.5% natamycin for fungal keratitis in Bangladesh. *British Journal of Ophthalmology* 1998;82(8):919-25. doi: 10.1136/bjo.82.8.919
11. Rahman MR, Minassian DC, Srinivasan M, et al. Trial of chlorhexidine gluconate for fungal corneal ulcers. *Ophthalmic Epidemiology* 1997;4(3):141-49. doi: 10.3109/09286589709115721
12. Dart JKG, Saw VPJ, Kilvington S. Acanthamoeba keratitis: diagnosis and treatment update 2009. *American journal of ophthalmology* 2009;148(4):487-99.e2. doi: 10.1016/j.ajo.2009.06.009
13. Kosrirukvongs P, Wanachiwanawin D, Visvesvara GS. Treatment of acanthamoeba keratitis with chlorhexidine. *Ophthalmology* 1999;106(4):798-802. doi: 10.1016/S0161-6420(99)90169-0

14. Seal D, Hay J, Kirkness C, et al. Successful medical therapy of Acanthamoeba keratitis with topical chlorhexidine and propamidine. *Eye* 1996;10 (Pt 4)(4):413-21. doi: 10.1038/eye.1996.92
15. Martin MJ, Rahman MR, Johnson GJ, et al. Mycotic keratitis: susceptibility to antiseptic agents. *International ophthalmology* 1995;19(5):299-302.
16. FlorCruz NV, Evans JR. Medical interventions for fungal keratitis. *The Cochrane database of systematic reviews* 2015;4:CD004241. doi: 10.1002/14651858.CD004241.pub4

17. Geffen N, Norman G, Kheradiya NS, et al. Chlorhexidine gluconate 0.02% as adjunct to primary treatment for corneal bacterial ulcers. *The Israel Medical Association journal : IMAJ* 2009;11(11):664-68.
18. Oakley CL, Vote BJ. Aqueous chlorhexidine is an effective alternative to povidone-iodine for intravitreal injection prophylaxis. *Acta Ophthalmologica* 2017;95(8):e794-e94. doi: 10.1111/aos.13340
19. Steinsapir KD, Woodward J. Comment on Chlorhexidine Keratitis. *Dermatologic surgery : official publication for American Society for Dermatologic Surgery [et al]* 2017;43(9):1180. doi: 10.1097/DSS.0000000000001172
20. Steinsapir KD, Woodward JA. Chlorhexidine Keratitis: Safety of Chlorhexidine as a Facial Antiseptic. *Dermatologic surgery : official publication for American Society for Dermatologic Surgery [et al]* 2017;43(1):1-6. doi: 10.1097/DSS.0000000000000822
21. Biesman B. Commentary on Chlorhexidine Keratitis. *Dermatologic Surgery* 2017;43(1):7-8. doi: 10.1097/DSS.0000000000000978
22. Humphrey S. Commentary on Chlorhexidine Keratitis. *Dermatologic Surgery* 2017;43(1):9-10. doi: 10.1097/DSS.0000000000000899
23. Burton MJ, Pithuwa J, Okello E, et al. Microbial Keratitis in East Africa: Why are the Outcomes so Poor? *Ophthalmic Epidemiology* 2011;18(4):158-63. doi: 10.3109/09286586.2011.595041
24. Schein OD. Evidence-Based Treatment of Fungal Keratitis. *JAMA Ophthalmology* 2016;134(12):1372-73. doi: 10.1001/jamaophthalmol.2016.4167

VERSION 2 – REVIEW

REVIEWER	Dr Catherine Fullwood Manchester University NHS Foundation Trust, UK
REVIEW RETURNED	19-Jun-2020

GENERAL COMMENTS	Thank you for the revision. Generally the protocol reads better. However there are still a few points: There are still some points where the term MK/FK are being used interchangeably. Page 9: Quality of life: Please add more details on the scoring for WHOQL-BREF Page 11, line 54: The phrase "2. Anaesthetic eye drops to anaesthetise the cornea before for procedures such as microbiology samples: Proxymetacaine 0.5% eye drop Minims (Bausch and Lomb, UK)." should not contain "for". Page 12, line 24: In the phrase "In the context of the trial this will be performed by the supervising consultant ophthalmologist" this should be replaced by "these" Page 13: In phrase "will be conducted in the same way described for the baseline assessment" Please add in "as described"
--

REVIEWER	Darren S J Ting Academic Ophthalmology, University of Nottingham, Nottingham, UK.
REVIEW RETURNED	03-Jul-2020

GENERAL COMMENTS	The authors have successfully addressed all the queries that were previously raised. Only two minor comments - I wonder if the authors could provide additional information in the manuscript itself (as they have already done in the revision letter):  1. In Page 5 “Objective” section, first line – Would be clearer to the readers if they could change “.... for treating fungal keratitis to “.... for treating filamentous fungal keratitis”. 2. Page 7 “Eligibility criteria” section – Please provide the specificity results of IVCM in this section (which the results have been provided in the revision letter) to support the rationale of using IVCM as the diagnostic criteria.
---

VERSION 2 – AUTHOR RESPONSE

Reviewer: 2

Thank you for the revision. Generally the protocol reads better. However there are still a few points:

There are still some points where the term MK/FK are being used interchangeably.

Thank you for highlighting this. I have clarified the instances where “microbial keratitis” should be “fungal keratitis”. Where we specifically mean microbial keratitis in terms of the introduction or patient flow / methodology, we have left this term in use.

Page 9: Quality of life: Please add more details on the scoring for WHOQL-BREF

We have included the scoring forms for all the Quality of Life tools as an online supplementary file, which include details of how this is scored.

Page 11, line 54: The phrase "2. Anaesthetic eye drops to anaesthetise the cornea before for procedures such as microbiology samples: Proxymetacaine 0.5% eye drop Minims (Bausch and Lomb, UK)." should not contain "for".

Thank you, this has been corrected.

Page 12, line 24: In the phrase "In the context of the trial this will be performed by the supervising consultant ophthalmologist" this should be replaced by "these"

Thank you, this has been corrected.

Page 13: In phrase "will be conducted in the same way described for the baseline assessment" Please add in "as described"

Thank you, this has been corrected.

Reviewer: 3

The authors have successfully addressed all the queries that were previously raised. Only two minor comments - I wonder if the authors could provide additional information in the

manuscript itself (as they have already done in the revision letter):

1. In Page 5 “Objective” section, first line – Would be clearer to the readers if they could change “.... for treating fungal keratitis to “.... for treating filamentous fungal keratitis”.

Thank you for this comment. This sentence now reads as follows:

The primary objective of this study is to determine if topical chlorhexidine 0.2% is non-inferior to topical natamycin 5% for treating filamentous fungal keratitis.

2. Page 7 “Eligibility criteria” section – Please provide the specificity results of IVCM in this section (which the results have been provided in the revision letter) to support the rationale of using IVCM as the diagnostic criteria.

Thank you for this comment. We have now added the following referenced sentence:

There is strong evidence supporting the use of IVCM for diagnosing filamentary fungal keratitis. Studies have reported sensitivities of 85.7 – 89.2% and specificities of between 81.4 and 92.7%, respectively. [1,2]

1. Chidambaram JD, Prajna NV, Larke NL, et al. Prospective Study of the Diagnostic Accuracy of the In Vivo Laser Scanning Confocal Microscope for Severe Microbial Keratitis. *Ophthalmology* 2016;123(11):2285-93. doi: 10.1016/j.ophtha.2016.07.009
2. Vaddavalli PK, Garg P, Sharma S, et al. Role of confocal microscopy in the diagnosis of fungal and acanthamoeba keratitis. *Ophthalmology* 2011;118(1):29-35. doi: 10.1016/j.ophtha.2010.05.018